# Clarithromycin-Loaded Submicron-Sized Carriers: Pharmacokinetics and Pharmacodynamic Evaluation

**DOI:** 10.3390/ma16093593

**Published:** 2023-05-08

**Authors:** Reetika Rawat, Raghuraj Singh Chouhan, Veera Sadhu, Manu Sharma

**Affiliations:** 1Department of Pharmacy, Banasthali Vidyapith, Banasthali 304022, Rajasthan, India; 2Department of Environmental Sciences, Institute “Jožef Stefan”, Jamova 39, 1000 Ljubljana, Slovenia; 3Centre for Advanced Materials Application, Slovak Academy of Sciences, Dubravskaesta 9, 84511 Bratislava, Slovakia; 4Polymer Institute, Slovak Academy of Sciences, Dúbravská Cesta 9, 84541 Bratislava, Slovakia

**Keywords:** *E. coli*, peritonitis, submicron dual lipid carriers, clarithromycin

## Abstract

The current study aims to improve clarithromycin bioavailability and effectiveness in complicated intra-abdominal infection management. Therefore, clarithromycin-loaded submicron dual lipid carriers (CLA-DLCs) were developed via hot high shear homogenization technique and evaluated for colloidal parameters, release behavior, stability study, and in-vitro antibiofilm activity. Bioavailability and therapeutic efficacy of optimized formulation on hampering cytokines storm induction was determined in *E. coli*-induced peritonitis. The developed CLA-DLCs (particle size 326.19 ± 24.14 nm, zeta potential −31.34 ± 2.81 mV, and entrapment efficiency 85.78 ± 4.01%) exhibited smooth spherical shapes and sustained in vitro release profiles. Long-term stability study of optimized CLA-DLCs ensured maintenance of colloidal parameters for 1 year at room temperature. In vitro antimicrobial studies revealed 3.43-fold higher anti-biofilm activity of CLA-DLCs compared with clarithromycin. In addition, the relative bioavailability of CLA-DLCs was enhanced 5.89-fold compared to pure drug in rats. The remarkable decrease in microbial burden in blood as well as tissues, along with oxidative stress markers (lipid peroxidation, myeloperoxidase activity, and carbonylated protein level) and immunological markers (total leukocyte count, neutrophil migration, NO, TNF-, and IL-6) on treatment with CLA-DLCs enhanced the survival in a rat model of peritonitis compared with the pure drug and untreated groups. In conclusion, CLA-DLCs hold promising potential in management of intra-abdominal infections and prevention of associated complications.

## 1. Introduction

Complicated intra-abdominal infections (cIAIs) are a complex fatal clinical challenge. The most alarming causative biofilm-forming pathogens, such as *Staphylococcus aureus*, *Haemophilus influenzae*, *Streptococcus pneumoniae*, and *Enterobacteriaceae* are of particular concern, as they are accountable for flare-ups of infections [1]. Thus, polymicrobial etiology and difficulty in diagnoses present an additional obstacle in cIAI management [2]. Furthermore, development of multidrug resistance makes cIAI management formidable and poses a significant burden on quality of life and the health-care system [3]. The associated complications, such as uncontrolled immune responses, leading to systemic inflammatory condition due to severe infection can lead to organ dysfunction, sepsis, or septic shock. Thus, peritonitis and coexisting alarming inflammatory responses are the vital cause of mortality in immunocompromised and critically ill patients [4]. Therefore, an urgent need exists to develop alternative antimicrobial therapy for management of cIAIs.

Several studies recommend macrolides alone or in combined antibiotic therapy as the treatment of choice due to their broad spectrum of antimicrobial activity [5,6]. Among macrolides, clarithromycin (CLA) is a drug of choice due to higher stability in acidic milieu of the stomach. It accumulates effectively in polymorphonuclear leukocytes and reaches the site of infection [7]. However, biopharmaceutical impedance, such as poor aqueous solubility and systemic bioavailability, extensive hepatic first pass metabolism, and higher dose frequency due to shorter half-life (3–5 h) limits its therapeutic efficacy in intracellular infections [8]. CLA also hampers hepatic microsomal cytochrome P3A4 required for its metabolism contributing to elevated serum levels of drug and hepatotoxicity [9]. Conventional formulations such as tablets and suspensions available in the market have experienced limited success in improving palatability, bioavailability, and therapeutic efficacy along with reduction of hepatotoxicity of CLA.

Recent expansions in nanotechnology have presented ample opportunities for efficient delivery of numerous therapeutic molecules [10]. The literature reports a variety of colloidal carriers, such as nano-emulsion [11,12], liposomes [13,14], cyclodextrin complexes [15], microspheres [16], lipid emulsion [17], hydrogels [18], polymeric nanoparticles [19,20,21,22], solid lipid nanoparticles [9,23], and niosomes [24] for topical, ocular, oral, and parenteral delivery of clarithromycin. However, most of the formulations documented are confined to either in vitro antimicrobial activity, permeability studies, or pharmacokinetic study, and they lack comprehensive details of anti-biofilm activity, pharmacokinetics, and pharmacodynamics. Furthermore, apprehensions related to inferior drug loading, biocompatibility, and scale-up have restricted their translation to clinical settings. In recent times, submicron-sized solid lipid carriers (LCs) have acquired intense recognition due to their substantial benefits, such as increased bioavailability, ability to retain structural integrity, and drug payload with high perspective for commercial scalability compared with liposomes and nano-emulsions [25]. LCs are composed of lipid matrices (e.g., waxes, fatty acids, and glycerides) having an inherent propensity to solidify at room temperature. The characteristic features, such as submicron size and lipid composition of LCs enhances their permeability across the gut wall by specific absorption mechanisms, such as endocytosis, pinocytosis, or M-cell-mediated transport, and bypasses portal circulation [26,27]. However, LCs composed of fatty acids or waxes usually exhibit higher crystalline structure, restricted drug-loading capacity, and drug expulsion during storage compared with triglycerides. The selection of a combination of fatty acids/ waxes and triglycerides can enhance physical stability and drug load due to blending of distinct constituents [28]. Thus, development of CLA-laden LCs using dual type of lipids could be an alternative strategy for preventing first-pass metabolism and to enhance the oral bioavailability of the drug [29]. Furthermore, lipophilicity of carrier systems can facilitate direct drug interaction to the bacterial cell wall and complex biofilms formed by pathogens and inhibit different stages of biofilm formation [9,30]. Lyophilization of LCs in powder form also enables loading into capsules, pellets, or tablets [27,31]. 

Therefore, the purpose of present investigation was to develop, characterize, and evaluate sustained-release CLA-loaded submicron dual lipid carriers (CLA-DLCs) for in vitro antimicrobial activity, bioavailability, and in vivo activity against *E. coli*-induced peritonitis in Wistar rats.

## 2. Materials and Methods

### 2.1. Materials

Clarithromycin was kindly provided as a gift sample by Sun Pharmaceutical Industries, India. Pluronic F 68 (PF-68), soya lecithin, stearic acid, compritol 888 ATO, and dialysis membrane was purchased from Hi-media (Mumbai, India). HPLC grade water, acetonitrile, and tween 80 was purchased from Merck, Germany. All chemicals were used as received without further purification. IL-6 and TNF-α kits were purchased from BD Bioscience (Minneapolis, MN, USA). Distilled water was utilized during the investigation.

### 2.2. Methods

#### 2.2.1. Preparation of CLA-DLCs

CLA-DLCs were fabricated using modified hot homogenization method [32]. Briefly, compritol 888 ATO and stearic acid were melted at 75 °C and mixed. Subsequently, CLA was dissolved in molten lipids to achieve oil phase. Simultaneously, aqueous phase was prepared by dissolving Pluronic F-68 (0.5% *w*/*v*) and modulated to temperature 70 ± 2 °C. Oil phase was subsequently added and emulsified under ultra-probe sonication for 10 min. The resulting emulsion was gradually cooled to room temperature, with continuous stirring. The obtained dispersion was centrifuged at 25,000 rpm at 4 °C for 20 min to collect particles. Supernatant was separated and put aside to evaluate encapsulation efficiency, as explained later. The resultant pellet was dispersed in distilled water and lyophilized at −55 °C and 0.07 bar pressure for 48 h using mannitol (5% *w*/*v*) as cryoprotectant. Lyophilized CLA-DLCs were stored in desiccator at room temperature. Various batches of CLA-DLCs were developed to optimize the various formulation and process variables are presented in Table 1. Each variable was optimized by performing the experiment in triplicate.

#### 2.2.2. Characterization of CLA-DLCs

##### Size, Polydispersity Index, and Zeta Potential

Lyophilized CLA-DLCs were analyzed for particle size, polydispersity index (PDI), and zeta potential utilizing dynamic laser light scattering spectroscopy (Nano ZS, Malvern, UK). The quantitative values of measurement parameters were as follows: measurement temperature of 25 °C, a medium viscosity of 0.8872, mPa s, and a medium refractive index of 1.330. Aqueous dispersion of lyophilized CLA-DLCs in distilled water was prepared by ultrasonicating samples for 30 s before evaluation.

##### Entrapment Efficiency

Supernatants collected after separation of CLA-DLCs pellet (as explained in preparation section) were estimated at 207 nm for CLA content using UV/Visible spectrophotometer. Entrapment efficiency was estimated by using the following equation [33]:(1)$$\;\mathrm{Entrapment}\;\mathrm{efficency}\;(\%)\;=\frac{Total\;amount\;of\;drug\;added-Free\;amount\;of\;drug}{Total\;amount\;of\;drug\;added}\times 100$$

##### Solid State Characterization

Surface morphology of optimized CLA-DLCs was analyzed utilizing scanning electron microscope (MIRA3 TESCAN). CLA-DLCs were mounted on aluminum stubs using double adhesive tape and sputter-coated with gold palladium alloy under an inert atmosphere of argon before capturing images at suitable magnification.

Attenuated total reflectance–Fourier transform infrared (ATR-FTIR) spectra of CLA, stearic acid, compritol, physical mixture of components, and optimized lyophilized formulation were acquired utilizing Bruker EQUINOX 55 FTIR spectrophotometer. Samples were inflexibly clenched in contrast to ATR lens and examined in transmission mode over wavelength range from 4000 to 600 cm^−1^.

Thermograms of CLA, stearic acid, compritol, physical mixture of formulation excipients, and optimized lyophilized formulation were recorded to determine drug–lipid compatibility by differential scanning calorimeter (DSC-204 F1 phoenix, Republic of Korea). Samples were placedin standard aluminum pans and heated at a pace of 10 °C min^−1^ over a range of 30–300 °C under continuous purging of nitrogen at 60 mL min^−1^.

X-ray diffraction patterns of CLA, stearic acid, compritol, physical mixture of drug and excipients, and optimized lyophilized CLA-DLCs were recorded by employing powder X-ray diffractometer (XPERT-PRO, PAN analytical, The Netherlands) with CuKα radiation (45 Kv, 40 mA) in the diffraction range of 0–80°, 2θ.

#### 2.2.3. Drug Release Studies

The assessment of CLA release from CLA-DLCs was determined in pH progressive media (HCl buffer pH 1.2 for 2 h, followed by phosphate buffer pH 6.8) utilizing dialysis bag diffusion method. CLA-DLCs equivalent to 10 mg of CLA dispersed in 1.5 mL distilled water was filled in dialysis bag (12–14 kDa) and immersed in dissolution medium (200 mL) kept at 37 ± 2 °C and 100 rpm. Sink condition was maintained by adding 0.1% *w*/*v* sodium lauryl sulfate. Sampling (3 mL) was executed at 0.5, 1, 2, 4, 6, 8, 12, 16, and 24 h, with consecutive replacement with equal volume of fresh buffer (37 °C). Samples were centrifuged, and supernatants collected were analyzed for CLA content. Drug release kinetics was determined by plotting obtained data according to different release kinetics models. The best fit model was determined on the basis of the numerical value of the calculated regression coefficient.

#### 2.2.4. Stability Studies

Optimized lyophilized CLA-DLCs stored in glass vials with polystyrene screw caps were examined for stability according to guidelines issued by ICH for Zones III and IV. Samples were kept at different storage conditions, including room temperature (25 ± 2 °C and 60 ± 5% relative humidity (RH)) and cool conditions (4 ± 2 °C and 65 ± 5% RH) for 12 months. Samples were evaluated for colloidal stability (particle size, PDI, and zeta potential) and entrapment efficiency at intervals of 0, 1.5, 3, 6, and 12 months.

#### 2.2.5. In Vitro Antimicrobial Activity

##### Minimum Inhibitory Concentration (MIC) Determination

The minimum amount of CLA and CLA-DLCs needed to inhibit growth of *E. coli* (MTCC code 433) was determined by broth dilution method. Concisely, sterile nutrient broth was inoculated with bacterial suspension (100 μL, equivalent to 0.5 McFarland standard) in culture tubes with subsequent addition of different concentration of test samples equivalent to 0.25, 0.50, 0.75, 1, 2, 4, and 6 µg/mL of CLA, respectively. The turbidity, an indication of bacterial growth in culture tubes, was analyzed at 600 nm after an incubation period of 24 h at 37 °C.

##### Minimum Biofilm Inhibitory Concentration (MBIC) Evaluation

Briefly, broth inoculated with *E. coli* (adjusted to 0.5 Mc Farland standards) for 24 h at 37 °C was decanted to remove non-adherent bacteria. Biofilms adhered to walls and bottoms of culture tubes were washed twice with sterile distilled water to draw out free bacteria. Subsequently, biofilm from the surface of tubes was detached, collected, and treated with various concentrations of CLA or CLA-DLCs equivalent to 25, 30, 35, 40, 70, 100, and 120 µg/mL of clarithromycin. Treated biofilms were withdrawn and incubated in fresh broth for 24 h at 37 °C. The presence of turbidity measured at 600 nm indicated presence of viable organisms [9].

#### 2.2.6. In Vivo Studies

Wistar rats (200 ± 20 g) acquired from the institutional animal house were kept on standard animal diet at 22 ± 2 °C and 55 ± 5% RH on a 12 h dark/light cycle. In vivo studies were approved by Institutional Animal Ethical Committee of Banasthali Vidyapith (574/GO/ReBi/S/02/CPCSEA), Rajasthan, India, and performed according to CPCSEA guideline established for care and use of laboratory animals.

#### 2.2.7. Pharmacokinetic Evaluation

Overnight-fasted Wistar rats were randomized into two groups (*n* = 6) for single dose oral bioavailability study. Animals of Groups I and II received CLA suspension (5 mg/kg) and CLA-DLCs (equivalent to 5 mg/kg of CLA), respectively. Blood samples (250 µL) were withdrawn from rat tail vein into heparinized tubes at 0.5, 1, 2, 4, 6, 8, 12, and 24 h postdosing. Blood loss was compensated by injecting equivalent volume of normal sterile saline solution. Later, blood was centrifuged at 5000 rpm for 10 min at 4 °C to segregate plasma and analyzed for drug content by HPLC method [9]. Pharmacokinetic parameters were estimated using WinNonlin software (5.1:Pharsight, Mountain View, CA, USA).

#### 2.2.8. Pharmacodynamic Studies

Intra-abdominal infection was induced via intraperitoneal injection of inoculum of *E. coli* suspension in normal saline (1 × 10^10^ CFU/mL). Subsequently, Wistar rats were randomized to receive CLA suspension (5 mg/kg), CLA-DLCs (equivalent to 5 mg/kg CLA), or normal saline orally (*n* = 6). Animals were euthanized after 24 h of respective treatment to collect lungs, liver, spleen, intestines, and blood samples via cardiac puncture aseptically. Blood samples were centrifuged at 5000 rpm and 4 °C to isolate plasma. Samples were evaluated for lipid peroxidation (LPO), carbonylated protein, myeloperoxidase (MPO) activity, nitric oxide (NO), SOD, catalase, GSH, TNF-α, and IL-6 level [34,35].

Microbial count in blood and tissue homogenates was determined by plating samples on agar plates, followed by incubation at 37 °C for 48 h and counting the colonies formed.

#### 2.2.9. Histopathological Evaluation

Isolated organs (lungs and spleen) were cut into small pieces of 0.5 cm^3^ and stored in 10% neutral buffered formalin. Samples were processed by serially dehydrating in increasing concentrations of ethanol, followed by clearing in xylene and embedded in paraffin using Leica EG 1160 Embedding machine. Transverse sections of 3 µm were cut, deparaffinized, hydrated to distilled water, and processed for hematoxyline–eosin staining. Tissue sections were observed under microscope, and images were captured at 40× magnification.

#### 2.2.10. Statistical Analysis

Data were presented as mean ± SD (*n* = 3). Statistical analysis using one-way ANOVA followed by Bonferroni’s multiple comparison test was performed utilizing GraphPad Prism version 5.00 (GraphPad Software, San Diego, CA, USA). *p* < 0.05 was considered as statistically significant in all studies.

## 3. Results and Discussion

### 3.1. Formulation Optimization

The remarkable variation in particle size and encapsulation efficiency was noticed on varying amount of lipids due to the differences in their HLB values. The average particle size and entrapment efficiency upon varying the compritol 888 ATO:stearic acid ratio from 1:2 to 2:1 ranged from 315.34 ± 24.15 nm to 579.24 ± 28.87 nm and from 61.64 ± 6.47% to 85.78 ± 4.01% (Table 2), respectively. The complex nature of compritol and its less perfect orientation leaves more space for drug loading in comparison with stearic acid. In addition, the long chain length of behenic acid in compritol 888 ATO enhances the intermolecular entrapment of the drug by interchain intercalation and intrusion in crystal lattice of stearic acid [36]. The enhanced CLA entrapment may be attributed to the fact that an increase in compritol 888 ATO content facilitates the reduction of the melting point of stearic acid and leads to massive disturbances in the lattice crystal order in stearic acid which bestows more space for drug loading in submicron lipid carriers [23,37] (Table 2). In addition, the relative increment of compritol content might have increased the viscosity of organic phase leading to faster solidification of submicron particles, which hindered diffusion of the CLA to the external phase [38]. However, an additional increment in compritol content of more than 50% *w*/*w* of total lipid had no improvement in CLA loading, manifesting saturation of the compritol effect [39]. Further increase in compritol content led to increase in particle size and PDI values, respectively along with decrease in entrapment efficiency (Table 2).

On increasing the pluronic F-68 concentration from 0.1% to 0.5% *w*/*v*, an increase in encapsulation efficiency and a decrease in particle size were noticed, respectively (Table 2). However, further increase in pluronic F-68 concentration (1% *w*/*v*) reduced size and entrapment efficiency (Table 2). The results established the need for notable amounts of surfactant to form stable colloidal systems and impede its solubilizing ability at higher concentrations [9,35,40]. The polydispersity index (PDI) of all batches prepared on varying surfactant concentrations and types of surfactants also followed a similar trend as particle size.

On varying the drug amount from 10 mg to 20 mg, an increase in encapsulation efficiency was observed. However, a further increase in drug amount (30 mg) led to a decrease in encapsulation efficiency due to a higher diffusion of the drug into the continuous phase as a relatively smaller proportion of lipid was present in comparison with drug amount.

An increment in external phase volume (50 mL) contributed to a decrease in encapsulation efficiency, with increase in particle size and PDI, respectively (Table 2). Insufficient amplitude of shear encountered at higher volume (50 mL) might have contributed to the decrease in encapsulation efficiency and rise of particle size and PDI, respectively [9,27].

An increment of sonication time from 7 min to 10 min had significantly lowered particle size and PDI along with improvement in encapsulation efficiency. However, further increment in duration of sonication (15 min) imparted higher shear stress, which might have ruptured the surfactant barrier and made the system thermodynamically unstable (Table 2) [27,41].

Zeta potential, a quantitative measurement of surface charge governs the physical stability of the system. Batches prepared on altering various formulation and process variables have zeta potentials in the range of −17.81 ± 2.19 mV to −31.34 ± 2.87 mV. Zeta potential of approximately ±30 mV generates columbic forces of higher magnitude, which overcome the van der Waals forces of attraction and represent higher physical stability of colloidal system [41].

Thus, on the basis of the outcomes, the CLA-DLC formulation (F2) produced utilizing compritol 888 ATO and stearic acid in a 1:1 ratio, drug amount of 20 mg, pluronic F-68 (0.5% *w*/*v*) as surfactant, 25 mL external phase volume, and sonication for 10 min was utilized for further studies.

### 3.2. Solid State Characterization

SEM microphotographs of CLA-DLCs demonstrated the formation of spherical-shaped submicron particles with smooth surfaces and uniform size distribution (Figure 1A).

The spectrum of clarithromycin displayed distinctive peaks at 2971.20 cm^−1^ of C-H stretching vibration, 1108.64 cm^−1^ and 1154.94 cm^−1^ due to ether functional band, C-C stretching between 800 cm^−1^ and 1200 cm^−1^, and peak at 1154.94 cm^−1^ representing C-N stretching along with methylene rocking vibration at 738.03 cm^−1^. The distinctive peak at 1689.45 cm^−1^, corresponding to -C=O stretching vibration of the carboxyl group, was observed in the stearic acid spectrum. The compritol spectrum exhibited characteristic peaks at 2912.69 cm^−1^ and 2846.83 cm^−1^ of C-H stretching, 1729.3 cm^−1^ and 1797.26 cm^−1^ refers to-C=O stretching, -C-O stretching carbonyl band refers to an ester band at 1174.33 cm^−1^, and methylene rocking vibration is seen at 1104.33 cm^−1^ and 716.47 cm^−1^. The spectrum of CLA-DLCs showed C-H stretching vibration at 2925.66 cm^−1^, -C=O stretching vibration at 1734.05 cm^−1^, C-O stretching vibration at 1087.94 cm^−1^ and 1022.48 cm^−1^, and methylene-group-associated vibrational band at 936.26 cm^−1^ and 880.66 cm^−1^ indicating the existence of clarithromycin in submicron lipid carriers deprived of any chemical interaction among excipients throughout CLA-DLCs preparation (Figure 1B). DSC thermogram of clarithromycin showed a sharp characteristic peak at 220 °C, while stearic acid and compritol exhibited broad peaks at 63.58 °C and 73 °C, respectively. Optimized formulation showed a broad endothermic peak at 80 °C. The non-existence of an endothermic peak of clarithromycin in the DSC thermogram confirmed the homogeneous dispersion of clarithromycin in optimized formulation (Figure 1C).

X-ray diffractogram of pure clarithromycin showed sharp and intense peaks at 9.40°, 11.40°, 12.32°, 13.67°, 15.11°, 17.22°, 18.98°, and 20.40° (2θ). Sharp diffraction peaks confirmed the crystalline nature of the drug [9]. The diffractogram of stearic acid exhibited broad halo peaks at diffraction angles of 21.36° and 25.81°. Similarly, the diffractogram of compritol showed broad halo peaks at diffraction angles of 21.16° and 29.95°. The physical mixture of clarithromycin with compritol and stearic acid exhibited distinctive peaks of clarithromycin in the diffractogram. However, the CLA-DLC diffractogram showed no existence of any distinctive diffraction patterns of clarithromycin in the optimized formulation indicating a uniform molecular dispersion of drug and loss of crystallinity of drug in LCs (Figure 1D).

### 3.3. Dissolution Study

Figure 2 represents the biphasic drug release pattern, i.e., an initial burst release (~16%, lasting 2 h in simulated gastric pH) followed by gradual drug release over 24 h (~90%) under bio-relevant conditions of the gastro-intestinal tract. Initial rapid release might have contributed to prompt diffusion and desorption of surface-adsorbed or weakly bound drug or partial film defects emerged during lyophilization [35]. The steady decline in the drug release rate might be ascribed to intrinsic drug solubility, path length of drug-depleted layers across which the drug travelled, rigidity of lipoidal barrier, and/or diffusion from dialysis membrane [27]. The results indicated that formulation would exhibit prolonged drug release behavior in simulated intestinal conditions.

Furthermore, a comparative evaluation of regression coefficients (R^2^) of zero order (R^2^ = 0.745), first order (R^2^ = 0.573), Higuchi model (R^2^ = 0.975,) and Korsmeyer–Peppas equation (R^2^ = 0.980) indicated best fit to Korsmeyer–Peppas equation. The value of the diffusion exponent (*n* = 0.81) also confirmed the anomalous drug release pattern of CLA-DLCs, i.e., dissolution and diffusion-controlled drug release across the lipid matrix [42].

### 3.4. Stability Studies

The F2 formulation stored at cool conditions and room temperature manifested no considerable alteration in particle size, PDI, zeta potential, or entrapment efficiency after 12 months of storage (*p* < 0.05) (Figure 3). This indicated that formulation variables were suitably optimized to retain pharmaceutical properties of CLA-DLCs for effective long-term use. Thus, results of stability study recommend storage of CLA-DLCs at room temperature conditions [27]. The physical appearance of CLA-DLCs did not show any noticeable changes at different environmental conditions.

### 3.5. In Vitro Anti-Bacterial Studies

Lower MIC value of CLA-DLCs (0.75 µg/mL) revealed higher effectiveness of formulation in inhibiting *E. coli* growth compared with free drug (6 µg/mL). Similarly, 3.43-fold lower MBIC of CLA-DLCs has assured its higher efficacy in inhibiting *E. coli* biofilm formation compared with the pure drug (Figure 4). The enhanced bactericidal activity of CLA-DLCs might have contributed to their ability to deliver their pay load into host cells through different mechanisms, e.g., contact release, adsorption, and endocytosis. The physiological resemblance of CLA-DLCs to membrane lipids might have improved uptake mechanism by fusion with cell wall or cell membrane. The enhanced cellular uptake and subsequent extended release of encapsulated CLA might have effectively improved the antibacterial effect of CLA-DLCs [9,43,44,45].

### 3.6. Pharmacokinetic Profile

Figure 5 represents plasma concentration versus time profiles for the single oral dose of CLA-DLCs formulation and CLA suspension, while corresponding pharmacokinetic parameters are presented in Table 3. CLA-DLCs attained 2.32-fold higher Cmax compared with CLA suspension. Submicron size as well as hydrophobic surface of CLA-DLCs might have aided their enhanced uptake and improved plasma drug concentration. Delayed Tmax, i.e., 4 h, also confirmed in vivo extended drug release profile of CLA-DLCs similar to in vitro release profile. CLA plasma concentration declined after 2 h, indicating rapid systemic clearance, which was further ascertained by lower MRT (Table 3). The elevated t_1/2_ (2.87-fold) and MRT (1.84-fold) of CLA-DLCs further confirmed the extended absorption and release of drug from CLA-DLCs in systemic circulation. The remarkable improvement in AUC_0–24h_ (*p* < 0.05) with CLA-DLCs (~5.93-fold) compared with CLA dispersion might be credited to improved systemic absorption of CLA-DLCs through paracellular and transcellular pathways via enterocytes, M-cells, and gut-associated lymphoid tissues or transport via triglyceride rich lipoproteins [33,35,46].

### 3.7. Pharmacodynamic Study

#### Effect of CLA-DLCs on Oxidative Stress Biomarkers

The quantitative results of bacterial cultures of blood and tissue homogenate samples of CLA-DLCs treated group showed remarkable reduction in microbial load (*p* < 0.05) compared with the pure drug and saline-treated groups (Table 4). An approximately 10^4^-fold lower microbial load in the blood, lung, and liver whereas 10^3^-fold change in spleen of CLA-DLC treated animals confirmed higher antimicrobial potential of CLA-DLCs compared with the pure drug.

The endotoxin administration manifested elevation of oxidative stress markers in lung, liver, and spleen (Figure 6). Pure drug and CLA-DLCs were found remarkably efficacious in bringing down the LPO, MPO, and carbonylated protein level in all tissues compared with the control (*p* < 0.05). The results revealed CLA-DLCs were more effective in lowering oxidative stress markers and bacterial count compared with the pure drug (*p* < 0.05) due to facilitated distribution of formulation in reticuloendothelial system rich organs, such as the lungs, liver, and spleen, major reservoirs of bacterial load, as is evident from pharmacokinetic studies [27,47] (Figure 6).

Significant elevation of serum total leukocyte count, neutrophils, lymphocytes, TNF-α, IL-6, and NO level in the control group on intraperitoneal administration of LPS was observed. Contrarily, treatment groups (CLA suspension and optimized formulation) showed a noticeable reduction (*p* < 0.05) (Figure 7). CLA ability to reduce circulating LPS by attenuating as well as killing bacteria might have contributed to a reduction in the level of inflammatory markers (TNF-α and IL-6) in plasma and infiltration of inflammatory markers (neutrophils and lymphocytes) in tissues [47,48]. CLA-DLCs treatment facilitated significant reduction in mean values of total leukocyte count, neutrophil, lymphocytes, TNF-α, IL-6, and NO level (*p* < 0.05) compared with drug and control (Figure 7). The unique characteristics of CLA-DLCs, i.e., submicron size, ease of adherence or diffusion to bacterial cell, and sustained drug release significantly inhibited bacterial growth and impeded the bacterial endotoxin-induced systemic inflammatory conditions. In addition, CLA-DLCs delivered higher CLA plasma concentrations upto 24 h and better distribution of CLA to tissues compared with the pure drug. Thus, CLA-DLCs significantly hampered the cascade of events induced by bacterial infection during peritonitis management.

### 3.8. Histology

Signs of lung injury—interstitial edema, alveolar thickening, and small peribronchial cellular aggregates with a large number of lymphocytes in interstitium and alveoliwere observed in lung tissues of the control group (Figure 8A). Histology of spleen tissues also demonstrated excessive infiltration of neutrophils, contributing to the disruption of normal architecture of the spleen (Figure 8D). The results of histology are in correlation with the level of oxidative markers and microbial count in respective tissues [49]. The endotoxins released and presence of *E. coli* in different tissues might have triggered the infiltration of neutrophils with elevation of MPO and LPO level, prominent mediators of tissue inflammation [48,49].

On the contrary, the CLA-treated group showed lower infiltration of neutrophils in tissues in comparison with the control (Figure 8B,E). The CLA-DLC-treated group showed significantly reduced neutrophil infiltration in the spleen and lung tissues compared with the drug and control groups (Figure 8C,F). Improved distribution and residence time of drug via CLA-DLCs might have provided the protection to tissues from damage by reducing microbial load, oxidative stress, and neutrophil migration [9,27,35].

## 4. Conclusions

CLA-DLCs were successfully fabricated using a hot homogenization method with higher entrapment efficiency and prolonged drug release behavior. Enhanced mean residence time and half-life of drug had remarkably enhanced bioavailability as well as tissue distribution. Formulation specifically reduced bacterial burden in organs, such as the lungs, liver, and spleen, where abscesses were present. CLA-DLCs have been found experimentally to be an ideal delivery system for mitigating the severity of infection by mopping up the endotoxins, preventing growth of *E. coli*, and reducing oxidative stress and tissue damage. Concisely, CLA-DLCs hold great promise as an effective therapy with the possibility of reduced dosing frequency for management of peritonitis and avoidance of further complications, such as septic shock.

## Figures and Tables

**Figure 1 materials-16-03593-f001:**
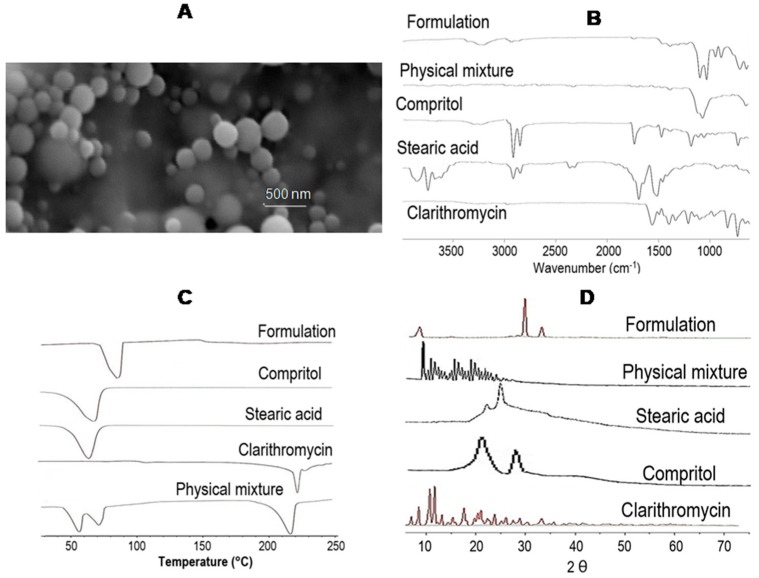
(**A**): Scanning electron microscopic images of optimized formulation; (**B**) Attenuated total reflectance-Fourier transform infrared spectra of clarithromycin, stearic acid, compritol, physical mixture, and optimized formulation; (**C**) Thermograms of clarithromycin, stearic acid, compritol, physical mixture, and optimized formulation; and (**D**) Diffractograms of clarithromycin, stearic acid, compritol, physical mixture, and optimized formulation.

**Figure 2 materials-16-03593-f002:**
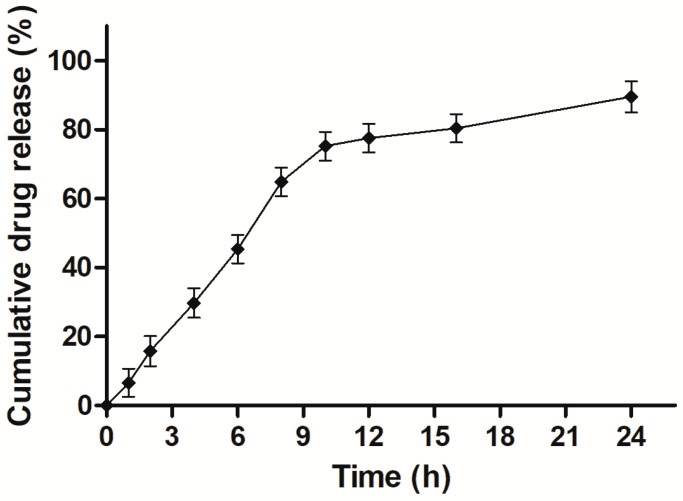
In vitro release behavior of optimized CLA-DLCs in pH progressive dissolution media. Results depicted are mean ± SD of triplicate experiments (*n* = 3).

**Figure 3 materials-16-03593-f003:**
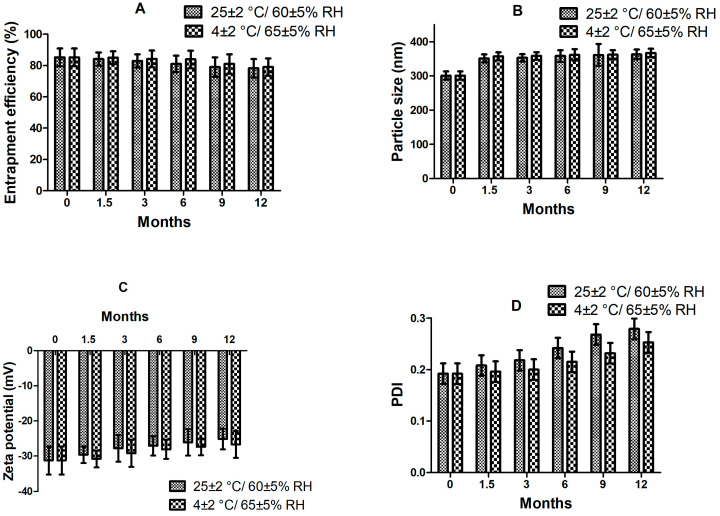
Effect of storage conditions on quality parameters: (**A**) entrapment efficiency, (**B**) particles size, (**C**) zeta potential, and (**D**) PDI of optimized formulation. Number of samples evaluated at each storage condition was three.

**Figure 4 materials-16-03593-f004:**
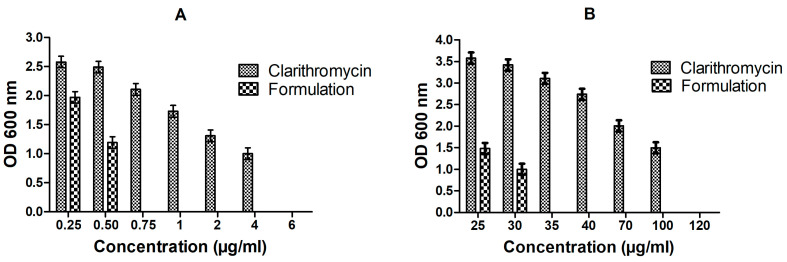
Inhibitory effect of clarithromycin and optimized formulation (CLA-DLCs) on (**A**) growth of *E. coli* (MIC determination) and (**B**) ability of biofilm to rejuvenate *E. coli* (MBIC assay) at different concentrations. Results illustrated are mean ± SD of triplicate experiments (*n* = 3).

**Figure 5 materials-16-03593-f005:**
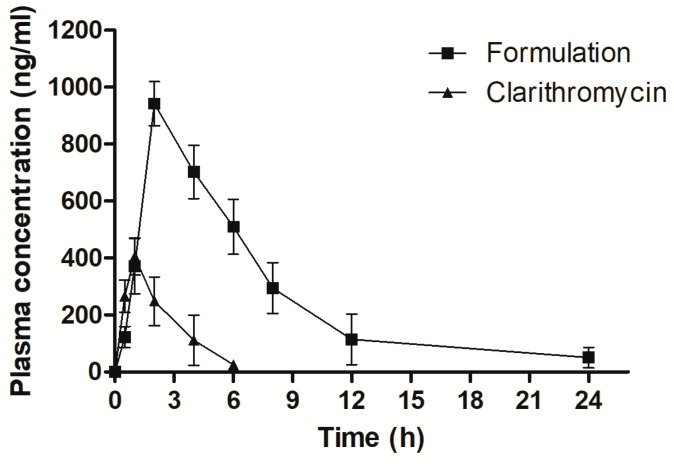
Plasma drug concentration versus time profile of clarithromycin and optimized formulation.

**Figure 6 materials-16-03593-f006:**
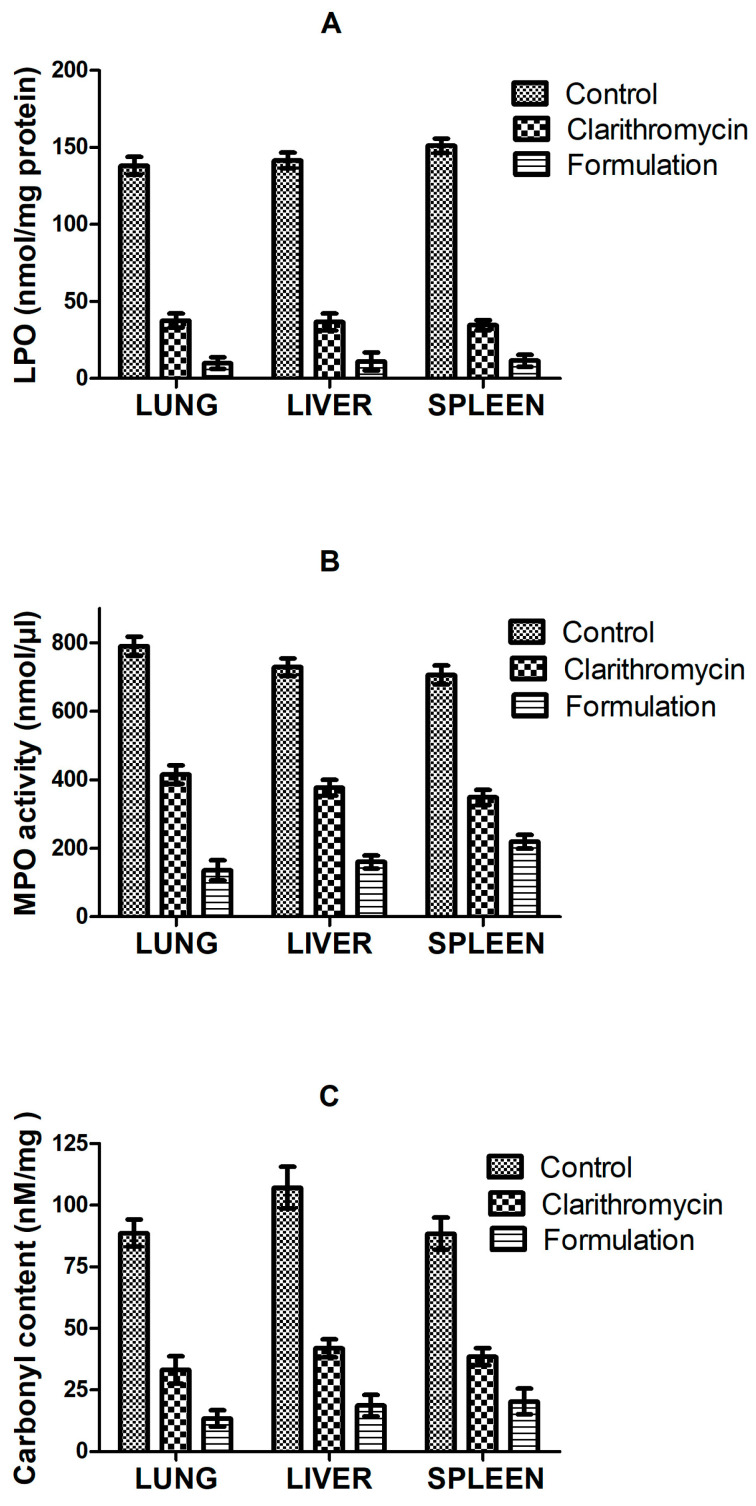
Effect of pure drug and optimized formulation against *E. coli* induced oxidative stress biomarkers: (**A**) lipid peroxidation (LPO), (**B**) myeloperoxidase (MPO) activity, and (**C**) carbonylated protein content. Results are expressed as mean ± SD, *n* = 6.

**Figure 7 materials-16-03593-f007:**
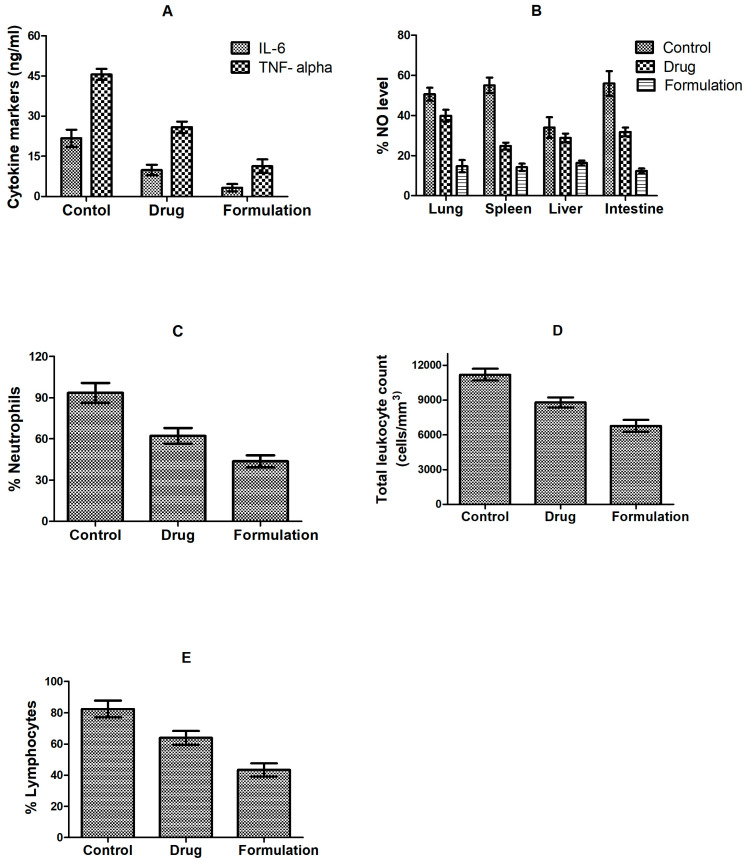
Effect of pure drug and optimized formulation after 24 h of treatment on level of immunological biomarkers: (**A**) IL-6 and TNF-α, (**B**) NO (%), (**C**) neutrophils count (%), (**D**) total leucocytes count; and (**E**) lymphocytes count (%) in *E. coli*-induced model of peritonitis. Results are expressed as mean ± SD, *n* = 6.

**Figure 8 materials-16-03593-f008:**
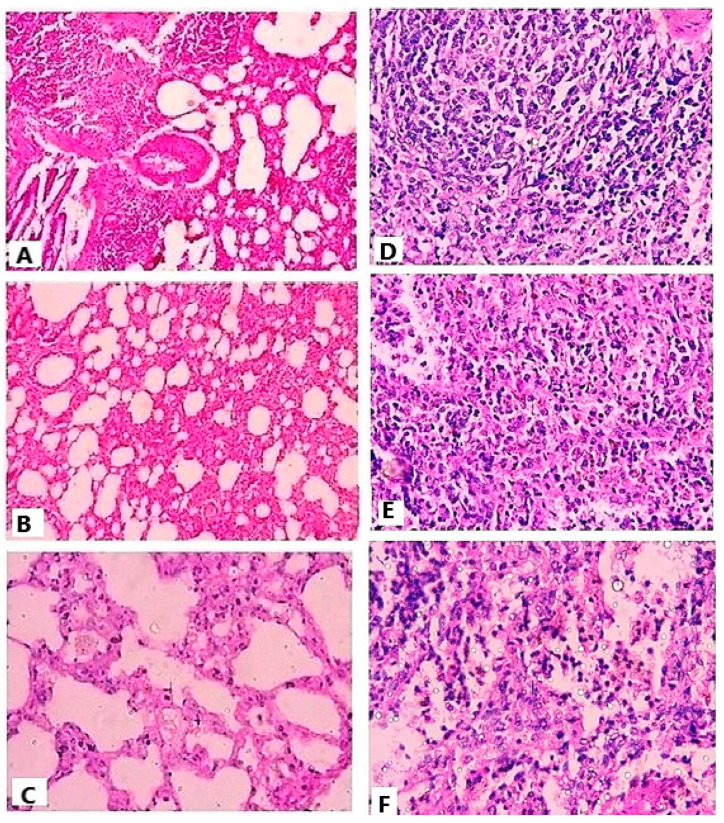
Histopathological images of lung and spleen of (**A**,**D**) control group, (**B**,**E**) pure-drug-treated group, and (**C**,**F**) optimized-formulation-treated group.

**Table 1 materials-16-03593-t001:** Various batches of CLA-DLCs prepared to optimize the variables.

F. Code	Compritol (mg)	Stearic Acid (mg)	PF-68 (% *w*/*v*)	Drug (mg)	Sonication Time (min)	External Phase Volume (mL)
F1	100	200	0.5	20	10	25
F2	100	100	0.5	20	10	25
F3	200	100	0.5	20	10	25
F4	100	100	0.5	20	10	50
F5	100	100	0.5	10	10	25
F6	100	100	0.5	30	10	25
F7	100	100	0.25	20	10	25
F8	100	100	1.0	20	10	25
F9	100	100	0.5	20	7	25
F10	100	100	0.5	20	15	25

**Table 2 materials-16-03593-t002:** Effect of diverse variables on quality parameters, i.e., particle size, polydispersibility index, zeta potential, and entrapment efficiency.

F. Code	Particle Size (nm ± SD)	PDI (PDI ± SD)	Zeta Potential (mV ± SD)	Entrapment Efficiency (% ± SD)
F1	316.21 ± 36.97	0.24 ± 0.02	−26.17 ± 2.14	74.87 ± 3.27
F2	326.19 ± 24.14	0.22 ± 0.04	−31.34 ± 2.87	85.78 ± 4.01
F3	424.54 ± 22.74	0.38 ± 0.03	−25.62 ± 2.25	81.03 ± 3.38
F4	506.17 ± 27.34	0.43 ± 0.02	−17.85 ± 2.14	65.35 ± 6.45
F5	315.34 ± 24.15	0.31 ± 0.03	−23.28 ± 2.94	75.27 ± 7.14
F6	392.43 ± 27.27	0.46 ± 0.04	−22.14 ± 2.15	77.31 ± 6.25
F7	501.14 ± 29.64	0.52 ± 0.03	−20.32 ± 1.96	75.78 ± 5.56
F8	329.29 ± 25.41	0.36± 0.04	−20.87 ± 2.83	71.94 ± 4.87
F9	579.24 ± 28.87	0.68 ± 0.04	−17.81 ± 2.19	61.64 ± 6.47
F10	464.42 ± 25.54	0.47 ± 0.03	−20.37 ± 3.15	71.01 ± 5.46

Values are expressed as mean ± SD, *n* = 3.

**Table 3 materials-16-03593-t003:** Pharmacokinetic parameters procured after single oral dose administration of formulation and clarithromycin suspension equivalent to 5 mg/kg^−1^ drug.

Parameter	Clarithromycin	Formulation
Cmax (ng mL^−1^)	405.59 ± 64.89	942.36 ± 77.71
Tmax (h)	1.00 ± 0.11	2.00 ± 0.13
Ke (h^−1^)	0.38 ± 0.07	0.12 ± 0.04
t_1/2_ (h)	1.38 ± 0.42	3.96 ± 0.67
MRT (h)	2.01 ± 0.53	3.69 ± 0.69
AUC (ngh^2mL^−1^^)	1058.67 ± 124.57	6278.47 ± 283.65

Values are expressed as mean ± SD, *n* = 3.

**Table 4 materials-16-03593-t004:** Effect of CLA and CLA-DLCs treatment on microbial count in blood and different tissues.

Treatment	CFU Count (Mean ± SD)			
	Blood	Lung	Liver	Spleen
*E. coli* (1 × 10^10^) (+ve control)	5.16 ± 0.84 × 10^10^	6.89 ± 0.56 × 10^8^	5.45 ± 0.25 × 10^9^	3.04 ± 0.75 × 10^8^
Pure drug (5 mg/kg)	4.17 ± 0.94 × 10^7^	5.65 ± 0.47 × 10^7^	5.27 ± 0.53 × 10^7^	1.69 ± 0.64 × 10^6^
CLA-DLCs (equivalent to 5 mg/kg)	3.28 ± 0.36 × 10^3^	4.45 ± 0.18 × 10^3^	5.62 ± 0.24 × 10^3^	2.14 ± 0.79 × 10^3^

## Data Availability

Submitted manuscript contains complete research data. No data has been shared in any repository.

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
