# Peer review of "Clarithromycin-Loaded Submicron-Sized Carriers: Pharmacokinetics and Pharmacodynamic Evaluation"

_materials, 2023, doi:10.3390/ma16093593_

Round 1
Reviewer 2 Report
Dear Authors;
Re: [Clarithromycin loaded nanostructured carriers: Pharmacokinetics and
Pharmacodynamic evaluation]
Manuscript ID: materials-2324169
Your research aim was to develop sustained release clarithromycin loaded nanostructured
lipid carriers (CLA-NLCs) with improved bioavailability and antimicrobial activity against E.
coli induced peritonitis. There are similar articles already available in the literature.
I am not sure about the novelty of your research reported here.
The manuscript was prepared somehow carelessly. Formatting was not consistent. Citations
and Referencing are done somehow in random. I highlighted few of these points in the
attached PDF file for your convenience.
Sections not done properly. For instance, section title:
"Preparation and characterization of CLA-NLCs" is not properly formatted.
There is no section title for "ENTRAPMENT EFFICIENCY" / "ENCAPSULATION
EFFICIENCY".
Table 1 contains research results, however, it is shown in the Materials and Methods section!
Figures do not have adequate resolution.
Figure legends need to be self-explanatory. If a reader decides to have a look at a data
without going through the text of your paper, need to get as much information as possible.
Hence, explain abbreviations in the Figure legends.
Figures' texts are too small to be legible.
Please explain what you mean by: "previously sanctioned by"? (in Line 171)
Animal studies are not mentioned in the Abstract. There is also no mention to the stability
results in the Abstract (this is a very important parameter in drug development).
n the section:
“Preparation and characterization of CLA-NLCs”
the exact temperature used must be specified.
The "n" number and evidence for the reproducibility of the procedure need also to be
mentioned clearly.
Results of particle size evaluation, PDI, ZP and EE must appear under the RESULTS section.
What was the quantitative value of particle sizing parameters: “Refractive Index” and
“Medium Viscosity”?
Number of experiments must be designated in all Tables and presented data.
Please provide Reference for equation 1.
Under the section: “Stability Studies” exact temperature must be given rather than “cool
conditions”
All abbreviations used must be defined when it is first mentioned in the manuscript.
Texts on Figure 1 B-D can hardly be seen.
Similar publications available in the literature include:
1. Sharaf M, Hamouda HI, Shabana S, Khan S, Arif M, Rozan HE, Abdalla M, Chi Z,
Liu C. Design of lipid-based nanocarrier for drug delivery has a double therapy for six
common pathogens eradication. Colloids and Surfaces A: Physicochemical and Engineering
Aspects. 2021 Sep 20;625:126662.
2. Xie S, Tao Y, Pan Y, Qu W, Cheng G, Huang L, Chen D, Wang X, Liu Z, Yuan Z.
Biodegradable nanoparticles for intracellular delivery of antimicrobial agents. Journal of
Controlled Release. 2014 Aug 10;187:101-17.
3. Moreno-Sastre M, Pastor M, Esquisabel A, Sans E, Viñas M, Fleischer A, Palomino
E, Bachiller D, Pedraz JL. Pulmonary delivery of tobramycin-loaded nanostructured lipid
carriers for Pseudomonas aeruginosa infections associated with cystic fibrosis. International
journal of pharmaceutics. 2016 Feb 10;498(1-2):263-73.
4. Kumari S, Goyal A, Sönmez Gürer E, Algın Yapar E, Garg M, Sood M, Sindhu RK.
Bioactive loaded novel nano-formulations for targeted drug delivery and their therapeutic
potential. Pharmaceutics. 2022 May 19;14(5):1091.
And several more articles.

Reviewer 3 Report
Line 155: please revise the grammar.
Line 216: can the authors explain how the compritol may fluidize the stearic acid chains? It is also possible that the combination of compritol with stearic acid generates a more amorphous structure that improves the CLA encapsulation efficiency. This phenomenon is reported and described in the literature.
According to SEM (Fig 1A) image, the nanoparticles population seems to be quite polydisperse, however, in DLS investigations it was noted a relatively good polydispersity of this batch of particles, for F2 (PDI = 0.22). Why there is this discrepancy?
DSC analyses (Fig 1C): the authors should revise the DSC results. There is a discrepancy between the values presented in thermograms of stearic acid and discussed in the main text. According to the literature, the melting point of stearic acid is approx. at 700C (as discussed in the main text), so, it is unclear why the thermogram with the endothermic peak at approx. 2200C is assigned to stearic acid.
Dissolution study: The authors should revise and improve the comments if this section. According to authors description in” materials and methods” section the release profile investigations were done in pH progressive media (HCl buffer pH 1.2 for 2 h followed by phosphate buffer pH 6.8), which is not reflect in the release profile discussion.
Stability studies: again, there is a discrepancy in the period during which the samples stability was investigated. What is the storage period? 6 months or 12 months?
In vitro anti -bacterial studies: Please expand what is MIC and MBIC. The authors attributed the increased antibacterial activity of CLA-NLCs to increased interaction of nanoformulated systems with E. coli membrane, but the particles are negatively charged as is the surface membrane of bacteria. Please explain the interaction mechanism, respectively the improved antibacterial effect.
Pharmacokinetic profile (Fig 5 and Table 2), please use the same codification of samples.
Line 155: please revise the grammar.
Line 216: can the authors explain how the compritol may fluidize the stearic acid chains? It is also possible that the combination of compritol with stearic acid generates a more amorphous structure that improves the CLA encapsulation efficiency. This phenomenon is reported and described in the literature.
According to SEM (Fig 1A) image, the nanoparticles population seems to be quite polydisperse, however, in DLS investigations it was noted a relatively good polydispersity of this batch of particles, for F2 (PDI = 0.22). Why there is this discrepancy?
DSC analyses (Fig 1C): the authors should revise the DSC results. There is a discrepancy between the values presented in thermograms of stearic acid and discussed in the main text. According to the literature, the melting point of stearic acid is approx. at 700C (as discussed in the main text), so, it is unclear why the thermogram with the endothermic peak at approx. 2200C is assigned to stearic acid.
Dissolution study: The authors should revise and improve the comments of this section. According to authors description in” materials and methods” section the release profile investigations were done in pH progressive media (HCl buffer pH 1.2 for 2 h followed by phosphate buffer pH 6.8), which is not reflect in the release profile discussion.
Stability studies: again, there is a discrepancy in the period during which the samples stability was investigated. What is the storage period? 6 months or 12 months?
In vitro anti -bacterial studies: Please expand what is MIC and MBIC. The authors attributed the increased antibacterial activity of CLA-NLCs to increased interaction of nanoformulated systems with E. coli membrane, but the particles are negatively charged as is the surface membrane of bacteria. Please explain the interaction mechanism, respectively the improved antibacterial effect.
Pharmacokinetic profile (Fig 5 and Table 2), please use the same codification of samples.
Round 2
Reviewer 2 Report
Dear Authors;
Re: Review Report, Round 2;
Manuscript Title: "Clarithromycin loaded submicron sized carriers: Pharmacokinetics and Pharmacodynamic evaluation"
Some of my comments / suggestions are not reflected in your revised manuscript. For example:
1. "I am not sure about the novelty of your research reported here."?
2. In the Abstract you mentioned:
"Stability studies recommended storage of CLA-LCs at room temperature CLA-LCs exhibited 3.43-fold higher anti-biofilm activity ...."
May I kindly ask authors, from the readers' point of view, is the a clearly written sentence in correct English?
3. Refractive Index and Medium viscosity are not reflected in the manuscript (you only added them to your answer list to my comments).
4. Some texts on Figures are still too small to be legible.
Reviewer 3 Report
The authors have done all the improvements recommended by the reviewer, the present form of manuscript can be accepted for publication.
Author Response
As per review report, reviewer has recommended acceptance of manuscript.